# Promoting Strategies for Healthy Environments in University Halls of Residence under Regular Epidemic Prevention and Control: An Importance—Performance Analysis from Zhejiang, China

**DOI:** 10.3390/ijerph192316014

**Published:** 2022-11-30

**Authors:** Yulu Zhao, Xinye Xu, Gangwei Cai, Zhetao Hu, Yan Hong

**Affiliations:** 1School of Civil Engineering and Architecture, Zhejiang Sci–Tech University, Hangzhou 310018, China; 2College of Civil Engineering and Architecture, Zhejiang University, Hangzhou 310058, China

**Keywords:** health promotion, university halls of residence, COVID-19, regular epidemic prevention, importance–performance analysis (IPA)

## Abstract

In the post-epidemic era, regular epidemic prevention and control is a daunting and ongoing task for nations all around the world. University halls of residence have been important spaces where university students balance their studies, work, and personal lives after COVID-19. Therefore, a healthy physical living environment deserves more attention. This paper compares situations before and after COVID-19 in an effort to evaluate the impact of indoor environments in university halls of residence on students. The study proposed eight vital dimensions for creating a healthy university hall of residence environment and, from 14 September to 4 October 2022, used an online questionnaire to collect data from 301 university students studying in Zhejiang, China. The key quality of service characteristics for fostering a healthy environment in university halls of residence were discovered using descriptive statistical analysis and revised importance–performance analysis (IPA). We found that an improved indoor physical environment and efficient arrangement of indoor space were crucial for the health of university students. The quality of educational services could be improved, and indoor exercise should be utilized effectively, both of which can contribute significantly to a healthy indoor environment. This study aims to contribute to the development of future initiatives to support healthy physical living environments in university halls of residence.

## 1. Introduction

As part of the future workforce, the health and well-being of university students have always been a top priority [1,2]. In February 2020, because of the global spread of COVID-19, China took steps to prohibit early return to study, implemented strict management of campuses, and offered distance learning courses to protect the lives and health of university students [3]. As the global COVID-19 epidemic persists, relevant prevention and control measures cannot be ignored, and China attaches great importance to them. China is still committed to the routine prevention and control of epidemics and places a high value on frequent monitoring and health protection on university campuses in order to provide students with a safe and healthy living, learning, and working environment [4].

Both the physical and mental health of university students have been impacted by COVID-19 [5,6]. One of the factors affecting their physical and mental health during the COVID-19 outbreak was discomfort in the indoor living environment [7]. Researchers are therefore working hard to identify effective interventions to enhance the living situations of students when they return to their studies in the wake of the COVID-19 outbreak. A significant proportion of previous research involves objective evaluations of the potential risk of virus transmission in the indoor physical living environment of university halls of residence and the protection of the health of university students [8,9]. Effective improvements to the indoor environment have successfully prevented the spread of viruses, but this places more emphasis on an objective assessment of the indoor physical living environment of university halls of residence. Fewer studies have sought to identify measures to improve the indoor physical living environment of university halls of residence from the perspective of their users.

University halls of residence are areas designated for use by a certain social group. University students not only reside here but also engage in behavioral activities such as studying and socializing. They are places where private and public spaces coexist, which is crucial for the development of the physical and mental health of university students [10]. Consequently, in light of regular epidemic prevention and control efforts and longer stays in halls of residence by university students, it is critical to investigate the many indoor environmental aspects that might enhance the health of the students, in addition to objectively evaluating the indoor physical living environment of the halls of residence. The need to protect the health of university students has become more pressing as a result of regular epidemic prevention and control efforts and the reopening of campuses throughout the world [11]. As a result, this study assesses several environmental factors in halls of residence that have an impact on the health of university students from the perspective of student satisfaction. In the main, these indoor environmental factors are physical living environment factors and do not include social and psychological variables, but we do consider the relationship between physical environmental factors and social psychological variables. The logical model for this investigation is depicted in Figure 1.

First, in order to create the questionnaire items, we reviewed a large number of studies to identify the elements of the indoor physical living environment that affect the health of university students. Second, from 14 September to 4 October 2022, 301 university students living in university halls of residence in Zhejiang Province, China, responded to surveys distributed via the social networking applications WeChat and QQ (Tencent, Shenzhen, China). Then, our study compared university students’ opinions of the health environment in their halls of residence before and after COVID-19 using descriptive statistics and revised importance–performance analysis (IPA). IPA is a diagnostic method that can help to identify the connection between importance and performance to aid future improvement initiatives [12]. Additionally, the elements influencing the quality of service for students in university halls of residence were determined in accordance with the findings of the revised IPA. Finally, certain supportive actions for the creation of a healthy environment in university halls of residence were also suggested in response to regular epidemic prevention and control.

The paper is structured as follows. The introduction is in Section 1. A survey of the literature on the study topic is presented in Section 2. The data collection and study technique are covered in Section 3. The findings of the study are presented in Section 4. The discussion and future research directions are set out in Section 5. Section 6 summarizes the study and draws conclusions.

## 2. Literature Review

### 2.1. From COVID-19 to Regular Epidemic Prevention and Control

COVID-19 poses a risk to public health. COVID-19 is still circulating today all over the world. In January 2020, the World Health Organization declared that COVID-19 was a public health emergency of international significance [13]. In April 2020, through the concerted efforts of the whole country, the spread of the epidemic in Wuhan was effectively controlled [14]. From that time, China actively promoted the orderly resumption of work and production while effectively controlling the epidemic and endeavoring to restore normality to everyday life and economic conditions [15]. Researchers also continue to make efforts to provide healthy and environmentally friendly living environments for individuals [16,17].

However, even with regular epidemic prevention and control, small-scale outbreaks of COVID-19 infection have occasionally occurred, and the crisis of localized outbreaks remains. Managing and controlling COVID-19 is significantly challenging because of the multiplicity of variables that may affect the propagation of the virus, such as population migration and air quality [18,19]. In addition, outbreaks disrupt the physical and mental health of individuals to varying degrees [20,21], and recovery from COVID-19 may still involve one or more persistent symptoms [22]. Therefore, it is impossible to overlook the long-term effects of the epidemic. A high value should be placed on stable measures to curb the spread of the virus and reduce the numerous negative effects of the epidemic on society and individuals, who must learn to adapt to this sustained policy of prevention and control.

### 2.2. Healthy Environments and Healthy Buildings

In 1948, when the World Health Organization was founded, health was defined as a state of complete physical, mental, and social well-being and not merely the absence of disease or infirmity [23]. Among the many aspects that affect human health, a healthy living environment is among the most crucial factors for enhancing health and happiness [24]. To ensure human health, creating a healthy living environment requires the cooperation of several policies, organizations, and sectors [25,26].

In order to provide individuals with healthy living environments, the building industry is constantly evolving [27]. Building project teams are able to minimize the environmental impact of buildings by using the Green Building Rating Systems (GBRSs), which provide a helpful framework and tools. LEED (US), BREEAM (UK), and CASBEE (JPA) are a few examples of green building rating systems which provide guidance for the construction of high-performance, sustainable structures [28,29]. Their development has contributed to the achievement of greater environmental and building sustainability. However, green building rating systems place more emphasis on the energy efficiency of buildings, from which most users do not derive benefit. A focus on user welfare and social advantages is required for green buildings to reach their full potential [30]. Currently, as individuals spend more time inside buildings, a condition known as “sick building syndrome” (SBS) has emerged. SBS can make people feel physically and psychologically uncomfortable and negatively affect their health [31]. As a result, there has been an increase in initiatives and attention to building-related human health problems. One example is the WELL Building Standard, developed by the International WELL Building Institute. The WELL Building Standard is an addendum that completes the scale for green building rating systems. In 2018, the WELL Building Standard V^2^ expanded the original seven concepts to ten [32]. Similarly, the Architectural Society of China has released guidelines for creating healthy environments [33]. This has considerably improved the indoor environments of buildings in all respects. Researchers are also increasingly examining the connection between the indoor physical living environment and mental health. According to one systematic review, there is also a significant relationship between specific housing qualities and the happiness of residents, which is crucial for both their health and well-being [34].

According to current studies, the primary focus of research to enhance the health of the built environment is the enhancement of physical elements such as air quality, light, heat, and sound in indoor environment quality (IEQ) [35,36,37]. Additionally, research has demonstrated that physical environmental factors of indoor environment quality (IEQ) not only have a significant impact on occupants’ perception of the building space but also affect their daily performance and productivity [38]. The management of the building and the services offered there are also related to the health of the occupants [39]. Furthermore, one of the aspects that is currently receiving attention is the promotion of healthy indoor activities [40]. The majority of these studies focus on a variety of building types, including apartments [41], office buildings [42], nursing homes [43], commercial buildings [44], and schools [45]. Among the types of school buildings, less attention has been paid to the health environment of university student dormitories. From the perspective of improving levels of satisfaction among university students, some scholars, including D. Amole [46], Thomsen and Eikemo [47], and Oke et al. [48], have offered recommendations for improving some of the functions of university halls of residence. They give particular attention to the distinctive spatial characteristics and facilities that are used daily by students in university halls of residence. However, the discussion of changes to promote the health of university students seems to have received less attention in the context of regular epidemic prevention and control.

### 2.3. The Impact of COVID-19 on the Health Environment of University Halls of Residence

The mental health and infection risk of university students have come under scrutiny because of COVID-19. The majority of students report that the COVID-19 outbreak has had a negative impact on their daily lives and academic performance [49,50]. Many campuses employ closed management practices, which restrict student movement and exacerbate psychological issues [51]. Additionally, they switched to remote learning as a method of education, and the academic performance of many university students suffered as a result [52]. Students who were isolated at home also frequently experienced symptoms of anxiety, sleep issues, higher levels of perceived stress, and stress related to distance learning [53,54].

Prior to readmitting university students to campus when COVID-19 subsides, campuses need to implement stringent behavioral interventions, comprehensively assess the campus environment for infection risk, and develop mitigation methods [55]. However, university students who must return to class also have concerns and face challenges in relation to their lives upon return. The worries of the students include the efficiency of the university’s outbreak prevention measures, the behavioral restrictions in force, living conditions in the dormitories, academic performance, and how often they are able to exercise [56]. Because of the limitations on movement around the university, students are likely to spend more time in their halls of residence each day. According to studies, students who share a bedroom are twice as likely to be infected as those who live alone [57]. In addition, many students find distance learning physically and psychologically stressful and are eager to return to school but reluctant to take the risk [58].

In view of the fact that the vast majority of universities have now reopened, it is reasonable to concentrate on supporting interventions that can enhance the health and well-being of university students once they return to university. However, research exploring how the improvement of the physical living environments of halls of residence impacts the health of university students is limited. Some studies have described the effects of the COVID-19 quarantine environment on university students’ mental health [59]. Others have carried out satisfaction surveys on the types of university halls of residence that find favor with students [60]. One research direction includes surveys of the thermal comfort of university halls of residence [61]. Few studies, however, have examined how satisfied university students are with the indoor physical living environment of their halls of residence under regular epidemic prevention and control compared to before COVID-19. As early as June 2020, university students in Zhejiang Province, China, a low-risk region for the disease, had already returned to class, and each university put stringent control measures in place [62]. In order to better understand how university students perceive the health environment in their halls of residence under regular epidemic prevention and control, we chose university students in Zhejiang Province, China, as the study population.

## 3. Materials and Methods

### 3.1. Data Collection

#### 3.1.1. Questionnaire Design

The purpose of this study was to investigate the perceptions and expectations of university students in Zhejiang Province, China, regarding the health of the physical living environments of their halls of residence and the implications of the association between their perceptions and expectations. The logic model for the questionnaire design is shown in Figure 2.

Firstly, the WELL Building Standard V^2^, which evolved from the Green Building Rating Systems, served as the basis for the design of the questionnaire [32]. Compared with the Green Building Rating Systems, the WELL Building Standard offers a collection of guidelines and strategies that support the consideration of occupant health and well-being in the planning and management of the indoor physical living environment of a building. Secondly, we also incorporated pertinent prior research findings to identify 8 key dimensions for creating a healthy environment in university halls of residence and sub-indicators for this study’s questionnaire because of the specificity of the student population at universities and the requirements of regular epidemic prevention and control. Additionally, an open-ended question was addressed to students who had been isolated in order to better identify the issues with indoor environments that concerned them most. Finally, the questionnaire was enhanced by combining the views of two professionals and four university students, resulting in a questionnaire with 26 sub-indicators (Table 1). A five-point Likert scale was used to evaluate the views of university students regarding the impact of COVID-19 on the health of the physical living environment in their halls of residence. There were five levels that could be selected in the questionnaire: (1) Importance (after COVID-19): “5 = very important”, “4 = important”, “3 = so-so”, “2 = unimportant”, and “1 = very unimportant”. (2) Performance (before COVID-19): “5 = very good”, “4 = good”, “3 = so-so”, “2 = not good”, and “1 = bad”.

#### 3.1.2. The Collection of Questionnaires

In order to gather feedback from university students in Zhejiang Province, China, this study used an online survey tool, SoJump (Changsha Ranxing Information Technology Co., Ltd., Changsha, China), to create the questionnaire and send it via WeChat and QQ (Tencent, Shenzhen, China), two of the most popular social networking services in China. The relevant datasets were then analyzed using SPSS. To ensure a broad pool of answers, the questionnaires were distributed at random to relevant student groups.

### 3.2. Importance—Performance Analysis (IPA)

#### 3.2.1. Concept of IPA 

Importance–performance analysis (IPA) is a basic evaluation and analysis technique. It was initially employed as a marketing tool to create marketing plans and organize tactical planning for increased market competitiveness [69,70]. IPA is now used in a variety of disciplines, including business management [70], healthcare [71], transport [72], education [73], tourism [74], digital media [75], and the construction industry [76,77]. Through the analysis of data, the primary goal of IPA is to diagnose the performance of various products or services and provide management with useful recommendations [78].

The traditional IPA analysis method investigates the importance and performance (satisfaction or service quality) of a product or service as perceived by respondents in the form of a scale, after which the data collected is statistically processed, and the mean value of the importance and performance of each question in the scale is used as a data point to create a 2-dimensional matrix. Figure 3 shows a 2-dimensional matrix of the performance and importance of the attributes perceived by the respondent based on IPA [69]. In this matrix, attribute importance is described along the *x*-axis, attribute performance (satisfaction or service quality) is described along the *y*-axis, and the matrix is divided into 4 quadrants. To enable managers to identify key characteristics, strengths, and shortcomings in a product or service and improve management methods, the quadrants of a 2-dimensional model each represent a particular strategy [79]. The attributes in Quadrant 1 (“Keep up the good work”) are considered major strengths and should be maintained or strengthened. The attributes in Quadrant 2 (“Possible overkill”) indicate inefficient use of resources. Managers can reduce their attention to these and redeploy resources where needed. The attributes in Quadrant 3 (“Low priority”) are considered to be relatively unimportant, secondary weaknesses, and low priority for management. The attributes in Quadrant 4 (“Concentrate here”) are the most critical and considered to be the main weaknesses. Managers need to give immediate attention and the highest priority to these in terms of resources and effort.

Therefore, this study is based on the traditional IPA analysis method. A diagnostic analysis of the perceptions of university students concerning the physical living environment in their halls of residence was used to identify areas of concern and improvement measures for university campus managers and policymakers.

#### 3.2.2. The Revised IPA Approach

Traditional IPA analysis is recognized as an effective analytical technique that requires data collection and analysis of both the respondent’s perceptions of attribute performance (satisfaction or service quality) and attribute importance dimensions [69]. The traditional IPA model is based on 2 assumptions: (1) that attribute performance and importance are both dependent variables, and (2) that the relationship between attribute performance and overall performance is linear and symmetrical [80]. A number of scholars have questioned the 2 assumptions of the traditional IPA model and have discussed and criticized them using proofs. They have suggested that changes in attribute performance (satisfaction or service quality) correlate with changes in attribute importance, findings that call into question the application of traditional IPA [81]. Using inappropriate methods to calculate performance or importance scores may result in incorrect or ineffective management strategies. Therefore, in the acquisition and analysis of importance attributes, scores can be obtained in 2 ways: (1) importance as stated by respondents and (2) implicitly derived importance obtained from some form of calculation.

In view of the limitations of the traditional IPA, as mentioned above, many scholars have revised and extended the importance score analysis of the traditional IPA method. Matzler et al. suggested that implicit importance could be obtained through partial correlation analysis between attribute-level performance and overall customer satisfaction [81]. In addition, Anderson et al. used multiple regression coefficients with natural log dummy variables as a measure of attribute importance [82]. However, the more widely used method for implicitly deriving the importance of attributes is that described by Deng. Based on a summary of previous research, Deng proposed a new method for implicitly derived importance: combining partial correlation analysis and natural logarithm transformation to calculate attribute importance [83]. The method consists of three steps:Step 1: Transform the performance of all attributes (AP) into a natural logarithmic form:
(1)$${\mathrm{AP}}_{i}\to \ln\left({\mathrm{AP}}_{i}\right)\;\;i\;=\;1,\;2,\;\ldots ,\;n$$
where n is the total number of attributes.

Step 2: Set natural logarithmic AP ($\ln\left({\mathrm{AP}}_{i}\right)$) and overall satisfaction (OS) as variables in a multivariate correlation model;Step 3: Execute partial correlation analysis for each attribute performance with OS. For example, if it is assumed that X_1_, X_2_, X_3_, X_4_, …, X_n_ are included in a multivariate correlation model, the coefficient of partial correlation between X_1_ and X_2_ when X_3_, X_4_, …, X_n_ are fixed is given by


(2)
$$\rho_{12\cdot 34\ldots n}\;=\;\frac{\sigma_{12\cdot 34\ldots n}\;}{\sigma_{1\cdot 34\ldots n}{\;\sigma }_{2\cdot 34\ldots \;n}}$$


Therefore, where OS is X_1_, $\ln\left({\mathrm{AP}}_{i}\right)$ is X_2_, and the rest of $\ln\left({\mathrm{AP}}_{i}\right)$ are X_3_ to X_n_; the partial correlation coefficient of the no. 1 attribute can be obtained using Formula (2).

The implicit derivation of attribute importance using Deng’s modified IPA method thus takes full account of the predictive validity of importance and optimizes it in the following three ways: (1) implicitly deriving importance eliminates the effect of correlation between attribute performance (AP) and overall satisfaction (OS); (2) biased correlation analysis eliminates multicollinearity between attribute variables; and (3) the natural logarithmic transformation captures the relevant attribute variables more sensitively [84].

Therefore, because traditional IPA studies do not adequately consider the predictive validity of self-stated absolute importance versus implicitly derived relative importance, this study used Deng’s revised IPA analysis method to further analyze the data to determine the impact of COVID-19 (before and after) on satisfaction with environments in university halls of residence from a health point of view.

## 4. Results

### 4.1. Descriptive Statistics

#### 4.1.1. Profile of Survey Respondents

The empirical data for this study came from a survey questionnaire distributed to most public and a small number of private colleges and universities in Zhejiang Province, China, between 14 September and 4 October 2022. The questionnaire received valid responses from a total of 315 respondents. As the target population for this study was residential students in Zhejiang Province, excluding students from outside Zhejiang Province and nonresidential students, 301 questionnaires were obtained after screening.

Table 2 describes the demographic profile of the respondents and the accommodation overview. Firstly, of the 301 university students, 40.20% were male (*n* = 121) and 59.80% were female (*n* = 180), and undergraduates and postgraduates accounted for 73.09% (*n* = 220) and 26.91% (*n* = 81), respectively. Secondly, the majority of university students lived in 3–4-bedroom halls of residence (70.43%, *n* = 212). In addition, on average, university students spent significantly more time in halls of residence on a daily basis after COVID-19 compared with before COVID-19. There was an increase in the proportion of students spending an average of 12–18 h per day in halls of residence (before COVID-19: 18.94%, *n* = 57; after COVID-19: 37.54%, *n* = 113) and a significant increase in the proportion spending an average of 18–24 h per day in halls of residence (before COVID-19: 3.32%, *n* = 10; after COVID-19: 14.95%, *n* = 45).

Finally, we assessed whether the respondents had experienced isolation and asked them an open-ended question: “What was the indoor environmental problem that was most detrimental to your health during the isolation period?” The statistics showed that 36.21% of university students had experienced isolation (*n* = 109). Twenty-one of these students provided feedback on the open-ended question, stating that they were troubled by poor air circulation, poor sound insulation, lack of space, lack of sunlight, dust, untimely removal of rubbish, inadequate or dilapidated equipment, and other problems during their time in isolation (Table 3).

#### 4.1.2. Reliability and Validity Analysis

Questionnaires were analyzed using SPSS 26 statistical software (IBM, New York, NY, USA). Internal consistency reliability tests are often conducted using Cronbach’s alpha [85]. Cronbach’s alpha values greater than 0.70 for each dimension were considered reliable [86]. The alpha for this questionnaire was 0.971 (alpha > 0.70), indicating relatively high and acceptable reliability. In addition, the questionnaire was further examined for construct validity, sample adequacy, and data fitness using the Kaiser–Meyer–Olkin (KMO) test [87]. When KMO > 0.70 and the *p*-value of Bartlett’s sphericity test is <0.05 (i.e., sig. < 0.05), the criteria are met. The questionnaire had a KMO of 0.972 (KMO > 0.70) and *p* = 0.000 (*p*-value < 0.05), indicating satisfactory construct validity (Table 4).

#### 4.1.3. Importance-Performance Scores

In order to compare management strategies between the traditional IPA and the revised IPA, the importance, as stated by respondents, was also collected using the questionnaire. The third and fourth columns of Table 5 list the data for self-reported importance and implicitly derived importance, respectively.

Using the IPA framework, the average response for attribute performance and implicitly derived importance of the 26 attributes was analyzed (Table 6). Variables in each category were ranked in order by paired differences (AP–IDI). The results showed that all data points (Sig. 2-tailed) were significantly below the 0.01 level, demonstrating that the variables were largely independent of one another, the data were spherically distributed, and the test results were acceptable and adequate.

### 4.2. Attribute Performance—Implicitly Derived Importance Analysis (Revised IPA)

Figure 4 shows the different analytical models obtained using the two different IPA methods described above. Different management strategies could be formulated according to the distribution of attribute satisfaction and importance. According to Figure 4b, i.e., using the modified IPA, the ranking by attribute number (QN) and the distribution of the 26 attributes in the obtained two-dimensional matrix is shown in Table 7.

## 5. Discussion

Because of the effects of COVID-19, more attention has been given to healthy buildings, but the health environment in university student halls of residence has received less attention. By examining university students’ expectations of the health environment in their halls of residence before and after COVID-19 using the revised IPA method, this study expands on previous research in the context of regular epidemic prevention and control. Understanding the management strategies related to the improvement of the indoor environment of university halls of residence could be very beneficial for campus administrators and policymakers and will be highly applicable for future optimization and enhancement. The findings indicated that there was a degree of discrepancy between university students’ expectations of the health environment in their halls of residence and actual performance under regular epidemic prevention and control. This was because of the significantly increased amount of time spent by students in their halls of residence, with a large number of attributes performing below their expectations. This leads to the following three conclusions.

Firstly, after using the revised IPA analysis, the installation of a fresh air system unit, a designated restorative space to support relaxation and rejuvenation, and an indoor exercise function moved from Quadrant 3 to Quadrant 4. Satisfaction levels remained low, but the importance scores were higher and demanded the attention of the relevant departments. In addition, the sanitary conditions of the toilet and bathroom, the indoor storage space, and wireless internet services for study and work consistently showed lower performance and higher expectations. University students who had been placed in quarantine provided responses that reflected similar worries. Students at universities generally expressed their discontent with poor indoor air quality and bathroom odor issues. Individual students reported that isolation damaged their mood and prevented them from exercising.

Secondly, in Quadrant 3, sound insulation of adjacent rooms and comfortable and private personal space had low performance, and the expectations of university students were also low for these attributes. Similarly, university students were less satisfied and less concerned regarding the use of natural materials and real plants in interior design and pleasant natural outdoor views from balconies. In addition, the public space exercise area inside the building had the lowest satisfaction rating when compared to indoor exercise and was less appealing to university students. We discovered that among the responses from students who had been placed in quarantine, a small number still felt that the isolation space was inadequate, and some thought that the sound insulation of the room was inadequate and would have liked more indoor greening. Even though there is no urgent need to improve these characteristics, the relevant departments should not ignore them.

Thirdly, in Quadrant 1 and Quadrant 2, regardless of whether revised IPA analysis was used, the majority of the indoor physical environments and epidemic-related services performed well. In general, with the exception of the factors in Quadrant 4 that required improvement, the performance of the air quality, sound environment, and water was good, with the lighting and thermal comfort of dorm rooms demonstrating the best performance. According to many previous studies, indoor environmental quality (IEQ) improvement was given a high priority by university campus managers as a result of COVID-19, and this is borne out by the results of our study [55,88]. However, there were still a small number of students who had been placed in quarantine and thought that some factors performed poorly. In addition, in Quadrant 1, university students regarded the comfort of tables and chairs and outside noise levels as being quite important. However, the comfort of tables and chairs and the provision of adequate natural sunlight in work areas were at risk of falling into Quadrant 4.

### 5.1. Implications for Theory

The main purpose of this study was to compare the effects of the indoor physical living environment in university halls of residence on the physical and mental health of students before and after COVID-19. Because of the recurrence and persistence of COVID-19, researchers are paying increasing attention to the health of university students, a susceptible population. We used the modified IPA analysis approach as the foundation for the study in order to better understand how the indoor physical living environment of university halls of residence affected the health of students from the perspective of the users. The modified IPA can help relevant departments to make defensible decisions that will maximize the satisfaction of university students and provide them with a better living environment while still adhering to regular epidemic prevention and control. IPA can effectively represent in a graphical form how university students perceive the indoor physical living environment in their halls of residence. Although IPA is a very useful technique, traditional IPA analyses have a number of significant flaws, in particular ignoring the relationship between changes in attribute importance and performance (satisfaction or service quality). Therefore, we adopted the modified IPA analysis method to make a better and more comprehensive judgment and analysis of the results. In addition, in order to address the drawbacks of the modified IPA, to perfect the usage of the IPA technique, and to give a useful theoretical explanation and practical test, our empirical comparison of the two IPA analysis methodologies highlighted discrepancies in the results. Overall, our work extends the use of modified IPA analysis in certain respects. It demonstrates that the application of the modified IPA technique may deliver more insightful and collaborative data to enhance the health benefits of the indoor physical living environment in university halls of residence under regular epidemic prevention.

### 5.2. Recommendations for Practice and Policy

As previously mentioned, university halls of residence have become crucial locations in which university students manage their studies, work, and personal lives since COVID-19. This study highlights several areas that require attention from university campus administrators and policymakers, as well as university students. The research aims to inform future efforts to develop healthy environments in university halls of residence.

The survey’s findings indicate that in order to meet the pressing needs of university students who must live, study, and work in halls of residence, university officials and decision-makers must improve the distribution of resources and provision of support. University students were eagerly anticipating the installation of new fresh air system equipment to improve the current interior air quality [89], as evidenced by the low satisfaction and high importance of this attribute (Quadrant 4) and comments from students who have been placed in quarantine. This might be connected to the condition of the toilets and the requirement for the relevant management department to promptly assess the toilet situation in their hall of residence and create an improvement plan. Additionally, even if a residence hall’s capacity is constrained as a building type for communal living, efforts should still be made to provide university students with some storage space [48]. Secondly, a designated space for relaxation and rejuvenation within the building is more crucial to the recovery of university students’ mental health than other strategies. Campus administrators and decision-makers should give more attention to the exercise of university students. University students considered that the indoor exercise facilities provided had poor performance, but they were considered relatively important. Related research has indicated that appropriate daily exercise during the epidemic could mitigate mental health problems. Administrators and policymakers should consider initiatives to encourage university students to exercise because this can help them to build stronger bodies and improve their mood, given that COVID-19 is a significant health-related concern [90,91]. Finally, it should be taken into consideration that, as part of the regular epidemic prevention and control measures, methods used to control the behavior of students can prevent them from meeting the demands of study and work in halls of residence. Better wireless internet connections are viewed as an urgent improvement issue for university students, which is closely related to their productivity [1].

In general, the factors in Quadrant 1 and Quadrant 2, with which university students expressed a high level of satisfaction and for which performance was better, should be maintained by campus administrators and decision-makers. It is important to note, too, that a small percentage of the university students who were placed in quarantine indicated in their feedback and responses to open questions that they were still bothered by interior lighting, sound insulation, and poor water supply. These factors play a very important role in the revised IPA chart. For instance, in future construction work, attention can be paid to adjusting the natural lighting of university students’ halls of residence so as to determine the visual comfort threshold for university students and provide more healthy natural lighting [92]. In addition, relevant studies show that the balcony, as a buffer space between the occupants and the outdoor green space, plays a critical function in helping to alleviate the mental health problems of university students and can help to improve their resilience [60]. Furthermore, the green visual ratio and areas of indoor and outdoor green space are also worth considering for improvement, as these are considered effective measures for enhancing resilience [93,94]. These two elements had a lower performance even though the study found that university students did not consider them very important. In order to ensure the healthy development of university students’ halls of residence in all respects, campus administrators and decision-makers should continuously work to provide a healthy indoor environment for students.

### 5.3. Limitations and Future Research Directions

This study is one of the first in China to reveal the satisfaction of university students with the health environment of their halls of residence in the context of regular epidemic prevention and control. The nature and breadth of this study have some limitations, notwithstanding the contributions it has made. Firstly, our study may not be indicative of a nationwide sample because we only surveyed university students in Zhejiang Province, China. Future research might include university students in other locations to produce more generalizable conclusions for reducing the health effects of COVID-19 on university students. Secondly, the WELL Building Standard V^2^ and earlier study findings provided the foundation for our eight key elements for a healthy environment in university halls of residence. A more in-depth analysis could be carried out in the future to determine the extent to which the relationship between these variables affects the development of a healthy environment in university halls of residence. Thirdly, based on the responses of quarantined students to the open-ended questions, several environmental characteristics that were not quantitatively examined in our study—such as garbage odor and outdated facilities—were also inferred. In addition, there are certain deviations according to individual preference regarding the indoor environment; for example, the importance of greenery varies from person to person. In the future, focus groups could be formed with subject-matter experts to actively explore additional qualities and facets of indoor environments that support the physical and mental health of university students, their perception of the effects of individual factors on the health environment of halls of residence, and to suggest strategies for improvement.

## 6. Conclusions

In this challenging period, although the COVID-19 outbreak is subsiding, the health of university students continues to cause concern. This study adds to the body of knowledge on the impact of COVID-19 on the physical and mental health of university students by contrasting conditions before and after COVID-19. It also uses the revised IPA to examine how university students perceive the performance of the health environment in their halls of residence and their expectations of it. Additionally, this study provides recommendations for practice and policy that can help university administrators and decision-makers enhance the health environment of university halls of residence under regular epidemic prevention and control measures. We recommend improving the indoor physical environment and arrangement of indoor space in university halls of residence, which can effectively promote a healthy environment. Furthermore, the focus should be placed on raising the bar for improving the standard of educational services and strengthening indoor exercise facilities, which are of great importance to the health of university students. In conclusion, in line with the development goal of various countries to create a healthy environment for university students, all stakeholders must work together to offer supporting interventions for the health and well-being of university students as the country’s future workforce.

## Figures and Tables

**Figure 1 ijerph-19-16014-f001:**
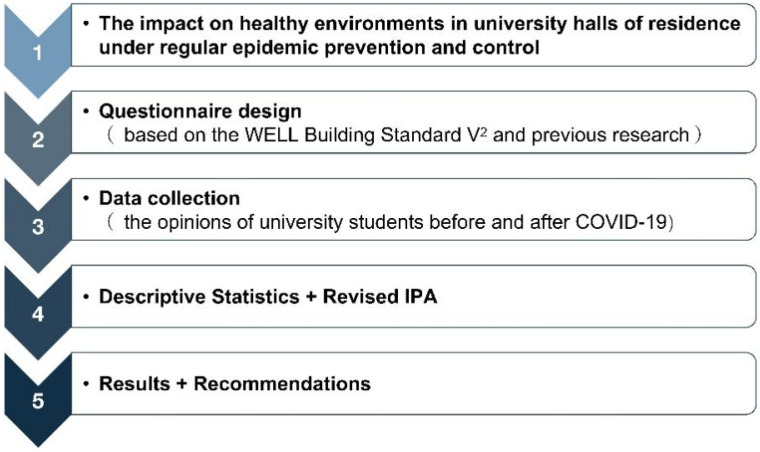
The logical model.

**Figure 2 ijerph-19-16014-f002:**
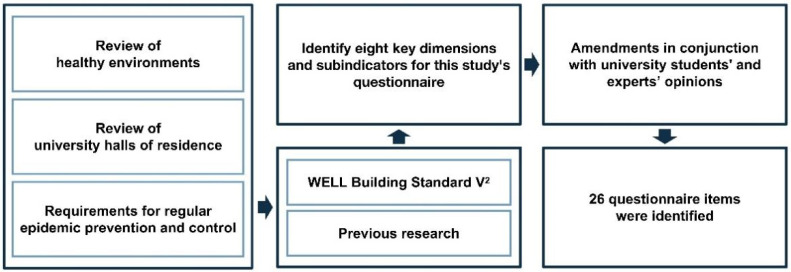
The design logic of the selection process of 26 research variable items.

**Figure 3 ijerph-19-16014-f003:**
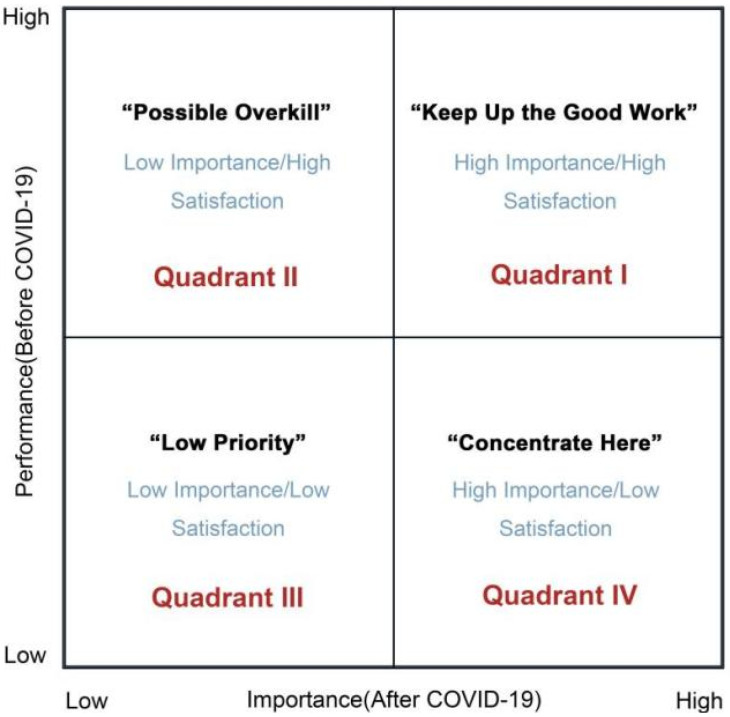
The importance–performance analysis (IPA) model.

**Figure 4 ijerph-19-16014-f004:**
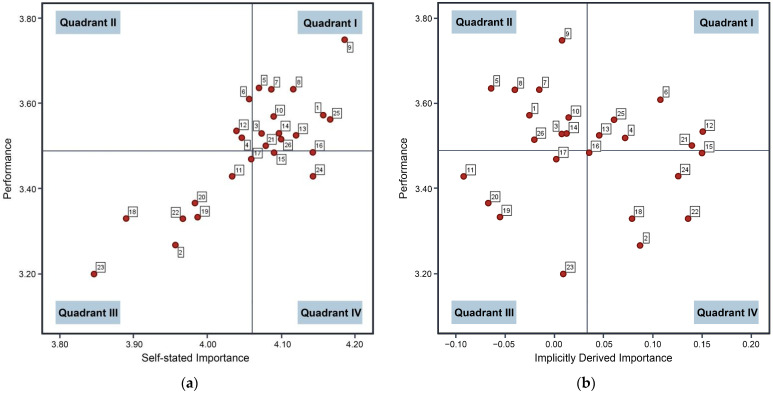
Analytical models for the traditional and revised IPA: (**a**) The performance–self-stated importance analysis model; (**b**) The performance–implicitly derived importance analysis model.

**Table 1 ijerph-19-16014-t001:** The 26 items for measurement of healthy environments in university halls of residence before and after COVID-19.

No.	Dimensions	26 Items: Importance (after COVID-19)/Performance (before COVID-19)	QN
1	Air Quality [32,63]	There is a sufficient number of openable windows for natural ventilation and fresh air.	1
		Air quality has improved with the installation of a fresh air system unit.	2
		No indoor odor (e.g., building material odor, tobacco odor).	3
2	Light [32,37]	There is natural sunlight in the room, and the work areas are well-lit and comfortable.	4
		There is no light in the visual field that is uncomfortable for the eyes (e.g., glare).	5
		Artificial lighting fixtures with light adjustment to comfortably meet the needs of day and night use.	6
3	Thermal Comfort [32]	The indoor temperature is comfortable.	7
		The humidity is appropriate, and the body feels fresh.	8
		The heating, fan, and air conditioning facilities are easy to use and comfortable.	9
4	Sound [32,64]	Interior noise exposure with acceptable levels (e.g., talking sound, HVAC).	10
		Walls and doors provide adequate sound isolation in adjacent rooms.	11
		Exterior noise exposure at acceptable levels (e.g., construction-related noise, traffic noise).	12
5	Water [32]	Drinking water of high quality; no contaminants.	13
		Convenient use of facilities for hand washing (e.g., hand sinks, soap boxes, hand dryers).	14
		Toilet and bathroom facilities meet needs while remaining clean and sanitary.	15
6	Space Perception and Mental Health [46,65]	Well laid out indoors with ample storage space.	16
		Personal space is comfortable, and privacy needs are met.	17
		One designated restorative space to support relaxation and rejuvenation within the building.	18
		Natural materials and indoor plants are inside to relieve fatigue and promote relaxation.	19
		Adequate balcony area with pleasant natural outdoor views (e.g., green and blue spaces).	20
7	Ergonomics and Movement [32,66,67]	Flexible and adjustable tables and chairs that can be used without discomfort.	21
		Indoor space can meet the requirements for basic exercise.	22
		A dedicated fitness facility and public space to support physical activity are available within the building.	23
8	Service and Management [1,55,68]	Good wireless internet connection to meet the needs of study and work.	24
		Hygiene services are provided in accordance with regulations (e.g., cleaning and disinfection).	25
		The residence hall management has a contingency plan for emergencies and promotes healthy lifestyles.	26

Note: No. = Dimension number, QN = Question number.

**Table 2 ijerph-19-16014-t002:** Profile of survey respondents (*n* = 301).

Variable		Number	Percentage
Gender	Male	121	40.20%
	Female	180	59.80%
Educational level	Undergraduate	220	73.09%
	Postgraduate	81	26.91%
Number of persons in halls of residence	1–2	13	4.32%
	3–4	212	70.43%
	5–6	69	22.92%
	7 and above	7	2.33%
Average daily time in halls of residence (before COVID-19)	0–6 h	42	13.95%
	6–12 h	192	63.79%
	12–18 h	57	18.94%
	18–24 h	10	3.32%
Average daily time in halls of residence (after COVID-19)	0–6 h	29	9.63%
	6–12 h	114	37.87%
	12–18 h	113	37.54%
	18–24 h	45	14.95%
Placed in quarantine	Yes	109	36.21%
	No	192	63.79%

**Table 3 ijerph-19-16014-t003:** Results of the open-ended question survey (*n* = 21).

Indoor Environmental Problems	Number (*n* = 21)
No air circulation; no fresh air	6
Poor sound insulation	3
Only one public toilet	2
Indoor space is insufficient	2
Garbage was not disposed of in time	2
Insufficient sunlight	2
Poor hygiene conditions	2
Dusty	2
Inadequate or dilapidated equipment (e.g., no washing machine, dilapidated air conditioning)	2
Fewer green plants	1
Unable to exercise	1
Inadequate water resources	1

**Table 4 ijerph-19-16014-t004:** Reliability and Validity Statistics.

Number of Attributes	Cronbach’s Alpha	KMO	*p*-Value
26AP + OS	0.971	0.972	0.000

Note: AP = Attribute Performance, OS = Overall Satisfaction.

**Table 5 ijerph-19-16014-t005:** Traditional IPA data and Revised IPA data (*n* = 301).

QN	AP	SSI	IDI	QN	AP	SSI	IDI
1	3.571	4.156	−0.025	14	3.528	4.096	0.012
2	3.266	3.957	0.087	15	3.485	4.090	0.150
3	3.528	4.073	0.008	16	3.485	4.143	0.035
4	3.518	4.047	0.072	17	3.468	4.060	0.002
5	3.635	4.070	−0.064	18	3.329	3.890	0.079
6	3.608	4.056	0.108	19	3.332	3.987	−0.055
7	3.631	4.086	−0.015	20	3.365	3.983	−0.067
8	3.631	4.116	−0.040	21	3.502	4.080	0.140
9	3.748	4.186	0.008	22	3.329	3.967	0.136
10	3.568	4.090	0.015	23	3.199	3.847	0.009
11	3.429	4.033	−0.092	24	3.429	4.143	0.126
12	3.535	4.040	0.151	25	3.561	4.166	0.061
13	3.525	4.120	0.046	26	3.515	4.100	−0.020

Note: SSI = Self-Stated Importance, IDI = Implicitly Derived Importance.

**Table 6 ijerph-19-16014-t006:** Rank means of attribute performance and implicitly derived importance and paired samples (*n* = 301).

Paired Differences (AP-IDI)				AP		IDI		Pearson Correlation	Sig. (2–Tailed)
QN	Mean	Rank	Std. Deviation	Mean	Rank	Mean	Rank		
9	2.487	1	0.713	3.748	1	0.008	16	0.968	0.000
5	2.406	2	0.697	3.635	2	−0.064	24	0.968	0.000
8	2.398	3	0.705	3.631	3	−0.040	22	0.969	0.000
7	2.396	4	0.696	3.631	3	−0.015	19	0.969	0.000
6	2.390	5	0.737	3.608	4	0.108	6	0.971	0.000
10	2.365	6	0.722	3.568	6	0.015	13	0.970	0.000
1	2.363	7	0.722	3.571	5	−0.025	21	0.969	0.000
25	2.362	8	0.732	3.561	7	0.061	10	0.969	0.000
12	2.349	9	0.761	3.535	8	0.151	1	0.969	0.000
13	2.342	10	0.732	3.525	11	0.046	11	0.971	0.000
3	2.339	11	0.754	3.528	9	0.008	17	0.970	0.000
4	2.336	12	0.738	3.518	12	0.072	9	0.970	0.000
14	2.334	13	0.716	3.528	10	0.012	14	0.969	0.000
26	2.330	14	0.733	3.515	13	−0.020	20	0.969	0.000
21	2.314	15	0.715	3.502	14	0.140	3	0.969	0.000
16	2.314	16	0.732	3.485	15	0.035	12	0.969	0.000
15	2.307	17	0.729	3.485	16	0.150	2	0.971	0.000
17	2.298	18	0.751	3.468	17	0.002	18	0.968	0.000
24	2.280	19	0.750	3.429	18	0.126	5	0.971	0.000
11	2.279	20	0.752	3.429	18	−0.092	26	0.970	0.000
20	2.238	21	0.765	3.365	19	−0.067	25	0.969	0.000
18	2.214	22	0.797	3.329	21	0.079	8	0.971	0.000
22	2.213	23	0.774	3.329	21	0.136	4	0.970	0.000
19	2.209	24	0.753	3.332	20	−0.055	23	0.971	0.000
2	2.167	25	0.735	3.266	22	0.087	7	0.969	0.000
23	2.145	26	0.824	3.199	23	0.009	15	0.972	0.000

**Table 7 ijerph-19-16014-t007:** Distribution of 26 attributes in revised IPA.

Quadrant	Quadrant 1 (Keep Up the Good Work)	Quadrant 2 (Possible Overkill)	Quadrant 3 (Low Priority)	Quadrant 4 (Concentrate Here)
Attribute Distribution (QN)	4	1	11	2
	6	3	17	15
	12	5	19	16
	13	7	20	18
	21	8	23	22
	25	9		24
		10		
		14		
		26		

## Data Availability

Not applicable.

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
