# Peer review of "Promoting Strategies for Healthy Environments in University Halls of Residence under Regular Epidemic Prevention and Control: An Importance—Performance Analysis from Zhejiang, China"

_ijerph, 2022, doi:10.3390/ijerph192316014_

Round 1
Reviewer 1 Report
Survey findings that dormitories and living conditions post COVID were inadequate in your sample of Chinese universities is important to share and may lead to important changes.
Your account of the pattern of the COVID epidemic in China and news about reduction of infections does not fit with international news coverage. Maybe cite studies to back up your presentation or correct what you say.
I thought you should have been more clear and direct that your study was about physical living conditions and that you would not include social psychological variables. That's OK, but the way you set up and justified the study had me looking for social psychological variables. I'd be more simple and direct, saying that the physical qualities of dormitory facilities is the focus. I think that's enough.
I thought your statistical presentation got in the way. You refer to methods I do not know about, and you did not explain how they work. My impression is that Chinese social science requires this over-emphasis on methods, but you need to explain both the method and how you got the results you did. In particular, when you show findings in terms of the four cell graphic, entries in the scattergram did not seem to match what you presented in the four cell statistical table. As a reader, I want to look at the data myself and come up with my own interpretation so I can match it with what you say. That did not seem possible since the statistical presentation and graphic presentation did not seem to match. It also would have helped to have you explain what is corrected between the first descriptive presentation of data and the second.
Author Response
Dear Editor and Reviewers: Thank you for your letter and for the reviewers' comments concerning our manuscript entitled "Promoting Strategies for Healthy Environments in University Halls of Residence under Regular Epidemic Prevention and Control: An Importance–Performance Analysis from Zhejiang, China" (ID: ijerph-2009916). The comments and suggestions are all valuable and very helpful for revising and improving our manuscript, as well as providing important guidance for our research. Through your precise revision comments, the quality of our manuscript has been significantly improved. We have studied the comments carefully and have made corrections, which we hope will meet with approval. The main corrections in the manuscript and the responses to the reviewer's comments are in the attachment. Please see the attachment.
Reviewer 2 Report
1. In this paper, the authors used ‘revised IPA’ as an approach to construct the basis of this research. However, the explanation of revised IPA is not clear enough. Please give more description.
2. As stated above, please explain how did authors transfer the number of revised IPA to make them significant and persuasive.
3. Comparing to other research related to IPA application, please specify your contribution and theoretical vision of revised IPA application.
Author Response
Dear Editor and Reviewers: Thank you for your letter and for the reviewers' comments concerning our manuscript entitled "The Promotion Strategies of Health Environments in University Residence Halls Under Regular Epidemic Prevention and Control: An Importance-Performance Analysis from Zhejiang, China" (ID: ijerph-2009916).Those comments are all valuable and very helpful for revising and improving our manuscript, as well as providing important guidance for our research. The quality of our manuscripts has been significantly improved by your precise revision comments. Thanks a lot for your considerate input – this is highly appreciated! We have taken all these comments and suggestions into account and made major corrections in this revised manuscript, which we hope will meet with approval. The main corrections in the manuscript and the responses to the reviewer's comments are in the attachment. Please see the attachment.
